# Pharmacological Inhibition of MMP-12 Exerts Protective Effects on Angiotensin II-Induced Abdominal Aortic Aneurysms in Apolipoprotein E-Deficient Mice

**DOI:** 10.3390/ijms25115809

**Published:** 2024-05-27

**Authors:** Karina Di Gregoli, Georgia Atkinson, Helen Williams, Sarah J. George, Jason L. Johnson

**Affiliations:** Laboratory of Cardiovascular Pathology, Department of Translational Health Sciences, Bristol Medical School, University of Bristol, Bristol BS2 8HW, UK; karina.digregoli@gmail.com (K.D.G.); do19070@bristol.ac.uk (G.A.); helen.williams@bristol.ac.uk (H.W.); s.j.george@bristol.ac.uk (S.J.G.)

**Keywords:** abdominal aortic aneurysm, matrix metalloproteinase, macrophage, pharmacology, proteomics

## Abstract

Human abdominal aortic aneurysms (AAAs) are characterized by increased activity of matrix metalloproteinases (MMP), including MMP-12, alongside macrophage accumulation and elastin degradation, in conjunction with superimposed atherosclerosis. Previous genetic ablation studies have proposed contradictory roles for MMP-12 in AAA development. In this study, we aimed to elucidate if pharmacological inhibition of MMP-12 activity with a phosphinic peptide inhibitor protects from AAA formation and progression in angiotensin (Ang) II-infused Apoe^−/−^ mice. Complimentary studies were conducted in a human ex vivo model of early aneurysm development. Administration of an MMP-12 inhibitor (RXP470.1) protected hypercholesterolemia Apoe^−/−^ mice from Ang II-induced AAA formation and rupture-related death, associated with diminished medial thinning and elastin fragmentation alongside increased collagen deposition. Proteomic analyses confirmed a beneficial effect of MMP-12 inhibition on extracellular matrix remodeling proteins combined with inflammatory pathways. Furthermore, RXP470.1 treatment of mice with pre-existing AAAs exerted beneficial effects as observed through suppressed aortic dilation and rupture, medial thinning, and elastin destruction. Our findings indicate that pharmacological inhibition of MMP-12 activity retards AAA progression and improves survival in mice providing proof-of-concept evidence to motivate translational work for MMP-12 inhibitor therapy in humans.

## 1. Introduction

Aortic diseases are diverse, occur in specific regions, and differ in their etiology. For instance, even though aneurysms can develop at numerous locations throughout the vascular tree, abdominal aortic aneurysms (AAAs) develop more commonly. It is estimated that up to 8% of men aged over 65 and up to 2.2% of women of the same age harbor aortic aneurysms [1]. Moreover, aneurysm rupture is estimated to be the tenth most common cause of mortality and accounts for 2% of all deaths [1]. Furthermore, while current repair and treatment strategies are effective in stabilizing large aneurysms (>50 mm in diameter), in-hospital mortality still represents a significant clinical problem [2]. Additionally, most AAAs detected by ultrasound are <50 mm in diameter, for which there is no standardized treatment. Hence there is a pressing requirement to identify medical therapies for the prevention of AAA progression.

Multiple pathological features characterize AAAs including compositional changes to key extracellular matrix (ECM) proteins such as collagen and elastin, alongside a reduction in vascular smooth muscle cell (VSMC) content through heightened apoptosis, and the increasing presence of neovascularization [3,4]. However, it is the accretion of inflammatory cells, especially macrophages, throughout all layers of the aortic wall, alongside resultant secretion of inflammatory growth factors, cytokines, and extracellular matrix (ECM)-degrading proteases that render AAAs distinctive from healthy aorta [5]. A similar accrual of inflammatory cells and mediators characterize atherosclerosis. Accordingly, studies elucidating the pathophysiology of AAAs have revealed that patients with AAAs frequently have atherosclerosis and associations have been shown by meta-analysis studies [6]. There are also numerous risk factors that are common with the pathogenesis of both pathologies, including smoking, hypertension, obesity, and age [6]. Relatedly, indicators of dysfunctional lipid metabolism (including raised circulating low density lipoprotein cholesterol (LDL-C) or lipoprotein(a) levels and systemic inflammation (such as interleukin (IL)-6 and interferon-γ (IFNγ) concentrations) are associated with the presentation of AAAs [7,8]. Genetic risk factors are also shared between AAAs and atherosclerosis [1] and a sequence variant on chromosome 9p21 is associated with atherosclerosis and aneurysms [9,10]. Moreover, intimal atherosclerosis is commonly present in AAA lesions [11], although the composition is different compared to coronary and carotid plaques, and medial elastin fragmentation is more prevalent [6]. Therefore, despite some subtle differences in composition compared with typical atherosclerotic lesions, AAAs are considered a form of atherosclerosis and are consequently regularly referred to as ‘atherosclerotic aneurysms’ [6,12,13].

Given that macrophage accumulation at both the adventitial and medial aspects is a striking feature of human atherosclerotic AAA [14], the principle detrimental role of macrophages in atherosclerotic AAA is considered to be through their release of proteolytic enzymes such as matrix metalloproteinases (MMPs) [15]. In turn, MMPs are apportioned responsibility for the excessive loss of ECM proteins observed within AAAs, especially elastin and fibrillar collagens, which characterize AAA progression and rupture [14]. MMPs are a family of proteases whose expression has been demonstrated to be abundantly expressed by macrophage infiltrates within AAAs [16]. Accordingly, MMP inhibition may be a potential therapeutic strategy for AAAs. However, a myriad of rodent studies has implied that inhibition of select MMPs which are detrimental is required, and necessary to obviate the abrogation of beneficial MMPs [16], as has been demonstrated for atherosclerosis [17,18,19]. Currently the inhibition of select MMPs in AAAs has not been assessed. MMP-12 is preferentially expressed by macrophages and facilitates monocyte/macrophage accumulation at sites of injury and during multiple inflammatory diseases such as atherosclerosis [20,21]. Although earlier genetic studies failed to identify an association for MMP-12 single nucleotide polymorphisms with AAAs [22], more recent genome-wide association studies have supported a genetic link between MMP-12 and AAA risk [23]. Increased MMP-12 expression is observed in human atherosclerotic and AAA samples relative to healthy aortae [24], and recent evidence has demonstrated restricted MMP-12 expression within a macrophage subset involved in atherosclerosis [25], indicating that MMP-12 may represent a valid therapeutic target to retard the formation and progression of atherosclerotic AAAs. However, MMP-12 gene deletion studies in multiple mouse models of AAA formation have provided inconsistent findings [26,27,28,29]. 

In the current study, we investigated the potential of deploying a pharmacological inhibitor of MMP-12 to suppress angiotensin (Ang) II-induced AAA development in Apoe^−/−^ mice. We revealed that the expression and activity of MMP-12 is augmented within atherosclerotic AAAs and demonstrated that inhibition of MMP-12 activity protects from both atherosclerotic AAA formation and progression through dampening of macrophage accumulation and associated elastin degradation.

## 2. Results

### 2.1. MMP-12 Is Elevated in Pro-Inflammatory Macrophages and Human AAAs, and Reduced MMP-12 Expression/Activity Retards Medial Fragmentation 

We have previously demonstrated that in vivo-generated rabbit foam cell macrophages display pro-inflammatory properties alongside increased MMP-12 gene and protein expression [26]. We now demonstrate increased expression of MMP-12 at both the mRNA (Figure 1A) and protein (Figure 1B) levels in human macrophages polarized to a pro-inflammatory phenotype through granulocyte–macrophage colony-stimulating factor (GM-CSF) differentiation compared to macrophage-colony stimulating factor (M-CSF) matured macrophages, which was independent of M1 (IFNγ + lipopolysaccharide (LPS)) or M2 (IL-4)) macrophage activation (Figure 1A). Moreover, elevated MMP-12 protein expression was identified within human AAAs compared with healthy aortic tissues (Figure 1C). Accordingly, MMP-12-positive macrophages were detected within human AAAs (Figure 1D) and associated with elastinolytic activity (Figure 1E). Concurrent in situ zymography in the presence of a selective MMP-12 inhibitor (RXP470.1) confirmed that MMP-12 activity is responsible for the majority of elastinolytic activity within human AAAs (Figure 1F,G). Comparably, atherosclerotic AAA tissues from Ang II-infused Apoe^−/−^ mice displayed medial elastin degradation (Figure 1H) which co-localized with elastinolytic activity (Figure 1I) that could be significantly reduced through inhibition of MMP-12 activity with RXP470.1 (Figure 1J,K). In line with these findings, a decreased incidence of medial elastin fragmentation was observed within atherosclerotic brachiocephalic arteries from Mmp12^−/−^ Apoe^−/−^ mice compared with Mmp12^+/+^ Apoe^−/−^ controls (Figure 1L). A similar protective effect on medial elastin fragmentation (MEF) was also detected within Apoe^−/−^ mice upon administration of RXP470.1 to selectively inhibit MMP-12 activity (Figure 1M). These observations suggested that dysregulated MMP-12 activity may be central to the development and progression of AAAs, characterized by elastin fragmentation and loss.

### 2.2. MMP-12 Inhibition Exerts Protective Effects on AAA Development

To directly determine the role of MMP-12 activity upon AAA formation, we treated Ang II-infused hypercholesterolemic Apoe^−/−^ mice with the phosphinic peptide RXP470.1, which is considered a selective inhibitor of MMP-12 activity [27] (Figure 2A). As expected with this model, 20% of Apoe^−/−^ mice suffered sudden AAA-related death, however this was significantly abrogated in RXP4701-treated mice as displayed through a 100% survival rate (Figure 2B). Moreover, histological analysis of AAAs showed a decrease in aneurysm severity in response to RXP470.1 treatment compared with untreated mice (Figure 2C), attributed to the absence of aortic ruptures (grade IV) within RXP4701-treated mice. In addition, although vessel perimeter, diameter, and area were unaffected with RXP470.1 treatment, the thickness of the media was significantly increased (Figure 2D,E) suggesting the stability of the vessel wall was augmented. In agreement, evaluation of the frequency in elastin breaks demonstrated RXP4701-treated mice exhibited a 40% reduction in elastin fragmentation (Figure 2F). Furthermore, although total vessel collagen content was unaffected between the two groups (Figure 2G), assessment of collagen fibers under polarized light revealed increased deposition of new collagen within the aortic wall of RXP470.1-treated Apoe^−/−^ mice (Figure 2H), further demonstrating MMP-12 inhibition increased stability upon the aortic wall. These findings suggest that inhibition of aberrant MMP-12 activity with RXP470.1 exerted protective effects on hypercholesterolemia and Ang II-induced AAA formation.

### 2.3. MMP-12 Inhibition Regulates ECM Remodeling Proteins and Inflammatory Pathways during Early AAA Formation 

Given the broad substrate repertoire of MMP-12 and potential effects on consequential signaling pathways, proteomic analysis of AAAs was undertaken to elucidate possible mechanisms underlying the protective effect of MMP-12 inhibition on Ang II-induced AAAs in Apoe^−/−^ mice. We identified 266 differentially expressed proteins (*p* < 0.05) between Apoe^−/−^ mice administered with RXP470.1 and untreated controls, divided into 65 upregulated and 201 downregulated proteins (Figure 3A). Interestingly, Saa1 and Saa2, acute phase proteins proposed to play causal roles in atherosclerosis and AAAs [28], were two of the most highly downregulated proteins within AAAs from RXP470.1-treated Apoe^−/−^ mice (Figure 3B). Conversely, two significantly upregulated proteins, Podxl and Aspn (Figure 3B), are suggested to exert anti-atherosclerotic and pro-fibrotic responses, respectively [29,30]. Differentially expressed proteins were explored within Ingenuity Pathway Analysis software to discern potential upstream regulators and identified 32 regulators with a significant number of their known targets (*p* < 0.05, activation Z-score > 2 or −2, containing at least six significantly changed targets) over-represented in the list of RXP470.1-modulated proteins whose collective differential expression indicated a change in activation status of upstream regulators (Figure 3C). Most of the regulators recognized as significantly activated or inhibited have been proposed to play pathophysiological roles in atherosclerosis and aneurysm progression. Specifically, two functional pathways were markedly affected in response to MMP-12 inhibition with RXP470.1, namely (1) regulators of inflammation (XBP1, IL5, IL6, TNF, TLR4, and REL), and (2) pathways involved in ECM remodeling (COL18A1, TGFB1, and THBS4). Notably, the AHR pathway displayed the greatest number of over-represented targets activated in AAAs from Ang II-infused Apoe^−/−^ mice with RXP470.1 administration (22 targets, *p* = 0.00176, Fisher’s exact test). Equally, the TP53 pathway had the most targets within the inhibited pathways (74 targets, *p* = 0.00055, Fisher’s exact test), suggesting MMP-12 inhibition affects cell survival within developing aneurysms. Consistent with an effect on immune cell regulation and ECM remodeling, enrichment analysis revealed multiple pertinent signaling pathways were significantly altered (false discovery rate (FDR) < 0.05) within AAAs from RXP470.1-treated Ang II-infused Apoe^−/−^ mice. Particularly, the immune system alongside collagen biosynthesis and modifying enzyme networks were enriched within analysis of biological processes (Figure 3D). Similarly, reactome pathway analysis identified multiple significantly enriched pathways associated with collagen and ECM organization (Figure 3E). 

These results suggest that inhibition of MMP-12 activity with RXP470.1 during AAA formation regulates a distinct protein profile which is required to mediate the inflammatory response and accumulation of extracellular proteins such as collagen and elastin, contributing to the protective effects on AAA development and related reduced incidence of sudden death.

### 2.4. MMP-12 Inhibition Alleviates the Progression of Pre-Existing AAAs 

To investigate the therapeutic potential of MMP-12 inhibition, we next investigated its ability to hinder the progression of pre-existing AAAs. Consequently, hypercholesterolemic male Apoe^−/−^ mice already subjected to four weeks of Ang II infusion (and therefore harboring AAAs), were treated for two weeks with RXP470.1 (Figure 4A). While 20% of control mice suffered sudden death through aortic rupture, all mice survived within the RXP470.1-treated group (Figure 4B). Histological evaluation revealed AAAs from RXP470.1-treated Apoe^−/−^ mice demonstrated reduced aneurysm severity, signifying progression was suppressed compared with control mice (Figure 4C). In accordance, AAAs were notably less dilated than those from controls as observed through reduced vessel perimeter, maximal diameter, and total vessel area with RXP470.1 treatment (Figure 4D,E). Moreover, two-week administration of RXP470.1 significantly reduced the number of medial elastin breaks within AAAs (Figure 4F), alongside augmenting the collagen content of AAAs, particularly deposition of new collagen (Figure 4G,H) indicative of a reparative response. Consequently, MMP-12 inhibition with RXP470.1 can also intervene upon the progression of pre-existing AAAs, altering morphological and compositional characteristics suggestive of healing advanced AAAs.

## 3. Discussion

The role of MMP-12 (also termed macrophage metalloelastase) has been explored in a multitude of cardiovascular diseases [16], with a pro-inflammatory and deleterious function demonstrated in atherosclerosis [30]. Moreover, studies utilizing mice with MMP-12-deficiency [31] or MMP-12 specific inhibition [21] to assess effects on atherosclerosis have reported reduced medial elastin destruction, implicating a role for MMP-12 in atherosclerotic aneurysm formation. Indeed, within the current study, we reveal for the first time that MMP-12 inhibition protected Apoe^−/−^ mice from atherosclerotic AAA formation, rupture, and death, observations, which were linked to changes in inflammatory and ECM remodeling processes, as corroborated through accompanying proteomic analysis. Demonstrating direct translational potential, MMP-12 inhibition suppressed the progression of pre-existing atherosclerotic AAAs.

Many studies have demonstrated an increased expression of MMPs in human and experimental mouse aortic dissection and aneurysm tissues, supporting a central role for dysregulated MMP expression and activity in the formation, progression, and rupture of aneurysms [32]. As its original name suggests (macrophage metalloelastase), MMP-12 is preferentially expressed by macrophages and plays a central role in monocyte/macrophage accumulation during multiple inflammatory diseases, such as atherosclerosis and pulmonary emphysema, predominantly through the degradation of elastin [20,21,33]. In line with our findings, elevated MMP-12 expression and activity is consistently observed within human and mouse atherosclerotic AAA tissues, with infiltrated macrophages displaying prominent expression, when compared with normal aortae [24]. As such, and due to its potent elastinolytic activity, numerous studies have explored the role of MMP-12 in the pathogenesis of AAs [34]. 

While our interventional findings utilizing pharmacological MMP-12 inhibition support a detrimental role for this protease, several studies evaluating the effect of MMP-12 deficiency in a range of mouse aneurysm models have yielded inconsistent findings [15]. Consistent with our own observations and supporting a detrimental role for MMP-12 in AAA, whole body deletion of MMP-12 reduced aneurysm growth elicited through adventitial CaCl_2_ application to wild-type mice, as evidenced by reduced elastin fragmentation and aortic dilation, in comparison to untreated control animals [26]. Also emulating our observations, a reduction in macrophage number was also reported within the media and adventitia, while levels of MMP-2 and MMP-9 were unaffected [26]. Protective effects of MMP-12 deletion were also reflected in AAAs induced by concurrent infusion of Ang II alongside a TGFβ neutralizing antibody in normocholesterolemic mice [28]. The genetic deletion of MMP-12 resulted in a reduced elastin degradation and aneurysm severity, culminating in protection from aneurysm rupture and related sudden death [28]. Further reinforcing an important role for macrophage-derived MMP-12 upon the adverse remodeling that characterizes AAs, Chen and colleagues [35] utilized transgenic rabbits, which express elevated MMP-12 levels in cells of a macrophage lineage. Employing an adventitial-applied carrageenan-induced model of AAA in hypercholesterolemic rabbits, the authors demonstrated heightened macrophage accumulation alongside augmented elastin degradation and aortic dilation in animals with macrophage MMP-12 over expression [35]. 

Predictably, given that MMP-12 is a potent elastase and consequently supports macrophage invasion through ECM remodeling, elastase-induced AAA development was unaffected by MMP-12 deletion in non-atherosclerotic mice upon a 129SvJ strain [27]. Meanwhile, a recent study proposed a protective role for MMP-12 in AAA formation and rupture, indicating that MMP-12 deletion in three differing mouse models (Ang II-infused Apoe^−/−^ mice, Ang II-infused C57Bl/6J mice with over-expression of a gain-of-function PCSK9 variant, and Apoe^−/−^ mice administered β-aminopropionitrile alongside perivascular porcine elastase application) accelerated death rates, which were attributed to accelerated aortic rupture [29]. Mechanistically, the authors proposed that macrophage-derived MMP-12 maintains vascular homeostasis through inactivation of the complement system and associated suppression of NETosis, as evidenced by augmented complement activity alongside neutrophil accumulation within AAAs of mice lacking MMP-12 [29]. It should be noted that superimposed atherosclerosis was absent within the aforementioned aneurysm studies. In addition to induction of atherosclerosis, hypercholesterolemia in Apoe-deficient mice induces monocytosis [36], which can be further amplified through Ang II-infusion and is associated with the development of AAA in Apoe^−/−^ mice [37]. Accordingly, Ang II-induced AAA formation and progression in Apoe^−/−^ mice is dependent upon the perpetual accumulation of monocyte-derived macrophages, which in turn is facilitated by MMP-12 expression and activity. Indeed, clodronate-induced monocyte depletion markedly reduced aneurysm rupture-related deaths in Ang II-infused mice treated with an anti-TGFβ antibody to induce macrophage MMP-12 expression, which was associated with a profound decrease in macrophage infiltration alongside preservation of elastin fibers [28]. Relatedly, we show that MMP-12 deficiency or inhibition limits medial elastin fragmentation within atherosclerotic brachiocephalic arteries. In support, Luttun and colleagues reported similar observations within the descending aortae of Mmp12^−/−^ Apoe^−/−^ mice, described as reduced atherosclerotic media destruction, which was accompanied by diminished elastin fragmentation, aortic dilation, and macrophage accumulation, independent of plaque number or size [31]. Taken together, these lines of evidence suggest a consistent deleterious effect of MMP-12 on aneurysm formation in the presence of atherosclerosis and accompanying macrophage-enriched inflammation, while in the absence of atherosclerosis and marked inflammation the role of MMP-12 is less clear. 

In divergence, Ang II-induced aortic aneurysm formation in C57Bl/6J mice is not associated with atherosclerosis, with aneurysm incidence and severity both markedly diminished, although total vessel and luminal diameters alongside medial wall thickness are increased [38]. These findings highlight that aortic aneurysm development in Ang II-infused wild-type mice occurs in the absence of atherosclerosis and/or hypercholesterolemia, indicating a principally VSMC-driven pathology as chronic inflammation is not observed. This mimics the pathophysiology of aortic dissections [39] and aligns with Ang II-infusion in wild-type mice serving as a model of aortic dissection, as opposed to AAA formation analogous to that in humans where atherosclerosis is present [40]. Further highlighting mechanistic differences between atherosclerotic and non-atherosclerotic aneurysms, a study from Salarian and colleagues [29] suggested a protective effect of MMP-12 upon AAA through the role as a negative regulator of complement activation. However, our proteomics analysis did not reveal any differential expression of complement system members with MMP-12 inhibition within mice harboring atherosclerotic AAAs. Notably, the majority of studies evaluating MMP-12 deletion in non-atherosclerotic mice report high mortality rates attributed to aortic dissection [29], as opposed to aneurysm formation and subsequent rupture. Indeed, combined in-depth histological and imaging studies in non-atherosclerotic mice (C57Bl/6J) have demonstrated that high-dose Ang II-infusion induces aortic dissection and associated sudden death during the first 0–7 days [40,41], which precedes AAA formation and inflammation [42]. Conversely, our data and others’ show death and early dissection rates are markedly lower in hypercholesterolemic Apoe^−/−^ mice, with atherosclerotic Apoe^−/−^ mice displaying large AAAs that are not observed in wild-type mice exposed to equivalent doses of Ang II [43]. Accordingly, it is plausible that MMP-12 can exert disparate effects during different phases of aneurysm formation and progression, especially in the absence/presence of atherosclerosis-related inflammation. 

Numerous studies deploying broad-spectrum MMP inhibitors in varying rodent models have demonstrated that indiscriminate metalloproteinase inhibition limits the development of AAAs [15]. However, such positive outcomes have not been replicated in clinical trials, which have focused upon tetracycline derivatives as broad-spectrum MMP inhibitors [15]. These disappointing results suggest that therapeutically targeting discrete MMPs may prove more effective. Adoption for this approach is supported by the observations that the modulation of individual endogenous MMP inhibitors, TIMPs, have yielded discordant effects upon AAA development. For example, TIMP-1 protects from elastin degradation/loss and aortic dilation in a mouse elastase-induced model of AAA [44], and a rat xenograft-induced AAA model [45]. However, studies utilizing the CaCl_2_-application mouse model revealed a detrimental effect of TIMP-2 upon aortic expansion, notwithstanding a favorable reduction in MMP-2 activity [46]. Conversely, a protective role for TIMP-3 on AAAs has been highlighted in both non-atherosclerotic and atherosclerotic mice Ang II-induced AAA mouse models, with both approaches demonstrating that Timp3-deficiency adversely affected vascular remodeling through increased inflammation and proteolytic activity, and consequently contributing to reduced collagen and elastin contents [47,48].

It is plausible that existing cardiovascular therapeutics utilized to treat associated risk factors or recommended lifestyle changes may directly alter MMP-12 levels and/or activity, and therefore be effective for the treatment of AAAs. While plasma levels of MMP-12 have been shown to be increased in individuals with type II diabetes mellitus, with elevated systolic blood pressure, or with increasing age [49], decreased circulating levels are observed in cardiovascular patients receiving anti-hypertensive drugs [50] or statin therapy [51]. Experimentally, the LDL-C-lowering drug atorvastatin was demonstrated to inhibit the development of AAAs in a rat elastase-induced model, an effect attributed to its anti-inflammatory actions, including suppression of macrophage accumulation and associated reduced MMP-12 expression [52]. Antioxidants have also been shown to exert protective effects on cardiovascular and metabolic diseases through anti-inflammatory actions, such as heme oxygenase-1 (HMOX1) [53]. Furthermore, HMOX1 can downregulate MMP-12 expression [54], and although not directly attributable to MMP-12, HMOX1 deficiency aggravated AAA development in Ang II-infused Apoe^−/−^ mice [55]. However, HMOX1 deficiency appears protective in normocholesterolemic mice, limiting AAA formation [56], similar to the proposed dual actions of MMP-12 [29].

In conclusion, our current findings confirm a deleterious role for aberrant MMP-12 activity in atherosclerotic aneurysm formation and rupture as demonstrated by the protective effect afforded on the formation and progression of established AAAs through MMP-12 inhibition with RXP470.1. Conversely, previous evidence suggests in the absence of atherosclerosis and associated inflammation, MMP-12 protects the integrity of the aortae and prevents acute dilation and subsequent aortic dissection. Therefore, we provided proof-of-principle evidence to support MMP-12 inhibitor therapy in patients with atherosclerotic AAA. However, caution should be taken when considering MMP-12 inhibition for the treatment of individuals harboring non-atherosclerotic dilatations, such as patients with bicuspid aortic valves and deemed at risk of ascending aortic dissection.

## 4. Materials and Methods

### 4.1. Human Aortic Tissue Samples and Cells 

Human AAA samples were collected from consenting patients undergoing aneurysm repair surgery, alongside non-aneurysmal aortic specimens under National Research Ethics Service approval (11/H0102/3). Peripheral blood mononuclear cells were isolated by differential centrifugation from whole blood of healthy donors, which were collected under South West 4 Research Ethics Committee reference 09/H0107/22. Monocytes were subsequently isolated by adherence and cultured in Roswell Park Memorial Institute (RPMI) media (ThermoFisher Scientific, Oxford, UK) supplemented with either 20 ng/mL recombinant human M-CSF (R&D systems, Abingdon, UK) or recombinant human GM-CSF (R&D systems, Abingdon, UK), for 7 days to generate mature macrophages. M0, M1, and M2 macrophages were polarized through the stimulation of mature macrophages (M-Mac and GM-Mac) with serum-free media, serum-free media with 20 ng/mL IFNγ (R&D systems, Abingdon, UK) and 100 ng/mL LPS (Sigma, Fareham, UK), or serum-free media with 20 ng/mL IL-4 (R&D systems, Abingdon, UK), respectively, for 24 h. Human aortic VSMCs were purchased from PromoCell and maintained per manufacture instructions with Smooth Muscle Growth Medium 2 (PromoCell, Heidelberg, Germany). For all experiments, the primary VSMCs were cultured for no more than four passages. 

### 4.2. Gene Expression

Total RNA was isolated from cell lysates within RLT buffer using a Rneasy Mini-Kit (Qiagen, Manchester, UK) according to the manufacturer’s protocol. RNA samples were quantified with a NanoDrop ND-1000 spectrophotometer (LabTech International, Heathfield, UK). The Quantitect Reverse Transcription Kit (Qiagen) was used to obtain equal amounts of cDNA from RNA samples; sample preparation and reaction mix was performed in accordance with the manufacturer’s instructions. The cDNA obtained was stored at −80 °C. Real-Time Quantitative PCR QuantiTect SYBR Green PCR Kit (Qiagen) was used to carry out quantitative PCR using a Roche LightCycler 480 (Roche, Welwyn Garden City, UK). Coding DNA was amplified using 4 ng of cDNA sample in accordance with the manufacturer’s instructions. QuantiTect pre-validated primer assays (Qiagen) were used for RPLP0 (NM_053275), ACTA2 (NM_001613), CALD1 (NM_033157), TAGLN (NM_001001522), MMP12 (NM_002426), and Mmp3 (NM_010809).

### 4.3. Western Blotting

Protein lysates were prepared in SDS lysis buffer and total protein concentration was measured using a bicinchoninic acid protein assay kit (ThermoFisher Scientific, Oxford, UK). Equal protein concentrations were loaded and electrophoresed on 4% to 12% gradient gels (Mini-PROTEAN TGX Stain-Free Precast Gels; Bio-Rad, Watford, UK) and transferred to 0.2 µm nitrocellulose membranes. Blots were blocked with 5% (*w*/*v*) skimmed milk powder and incubated overnight at 4 °C with rabbit anti-human MMP-12 (ab52897, Abcam, Cambridge, UK) or rabbit anti-human smooth muscle myosin heavy chain 11 (ab133567, Abcam), diluted 1/1000 in SignalBoost Solution 1 (Merck, Gillingham, UK). The primary antibody was detected using goat anti-rabbit horse-radish peroxidase (HRP)-conjugated secondary antibody (P0448, Agilent, Didcot, UK) diluted in SignalBoost Solution 2 (Merck) and enhanced Luminata Forte chemiluminescence reagent (Merck). The optical density of bands was quantified using the Gel Doc XR+ Gel Documentation System (Bio-Rad) and normalized to sample total protein content (whole lane) assessed using Bio-Rad stain-free technology, with the most intense band shown within representative figures and referred to as the loading control.

### 4.4. In Situ Zymography

Elastinolytic activity was localized in human and murine AAs using a previously described method [21]. In brief, frozen 8 μm cryostat sections were incubated overnight at room temperature in a humidified dark chamber with 20 μg/mL fluorescein (FAM)-conjugated elastin (Cambridge BioScience, Cambridge, UK) dissolved in developing buffer (50 mM Tris, pH 7.4, 150 mM NaCl, 5 mM CaCl_2_, 0.2 mM sodium azide). Cleavage of the substrate by elastinolytic proteases results in unblocking of quenched fluorescence and a concomitant increase in fluorescence intensity. The sections were incubated for 24 h in a developing buffer alone or in the presence of the MMP-12 inhibitor RXP470.1 (100 nmol/L). Sections were washed in PBS, fixed with 4% paraformaldehyde, and mounted with ProLong^®^ Gold antifade reagent with DAPI (Life Technologies Ltd., Paisley, UK). Using fluorescence microscopy, elastinolytic activity was identified as green fluorescence.

### 4.5. Mice

Apoe^−/−^ mice (JAX #002052) upon a C57BL/6J strain background were purchased from The Jackson Laboratory and maintained within our own animal facility. Animal numbers for each specific sub-study are provided within associated figure legends. The housing and care of the animals and all the procedures used in these studies were performed in accordance with the ethical guidelines and regulations of the University of Bristol and the UK Home Office. The investigation conforms to the *Guide for the Care and Use of Laboratory Animals* published by the US National Institutes of Health (NIH Publication No. 85–23, Eighth Edition, revised 2011). Adherence to the ARRIVE guidelines [57] for the reporting of animal in vivo experiments was also followed.

### 4.6. Mouse Model of Ang II-Induced AAA

To induce AAA formation in atherosclerotic plaque-prone Apoe^−/−^ mice, male animals at the age of 8–10 weeks were fed a high-fat diet containing 21% (*wt*:*wt*) pork lard and supplemented with 0.15% (*wt*:*wt*) cholesterol (Special Diet Services, Witham, UK) for 8 weeks, the last 3–4 weeks of which they were infused with Ang II, as previously described [47]. Briefly, Alzet model 1004 osmotic mini-pumps (Charles River, Canterbury, UK) were subcutaneously implanted to deliver Ang II (500 ng/kg/min, Enzo Life Sciences, Exeter, UK) for 21 or 28 days, depending on the experimental design. The mice were anaesthetized with isoflurane and oxygen (2.5% and 1 L/min), which was followed by mini-pump implantation. Post-operative analgesia was performed by intraperitoneal injection of buprenorphine (0.1 mg/kg) immediately after surgery, and the animals were closely monitored for 24 h and subsequent pain relief delivered if necessary. To ensure that the studies were adequately powered, group sizes of at least 8 animals were used per group, accepting a 15–20% attrition rate due to early aortic rupture and sudden death. To assess the incidence and severity of AAA formation, a method outlined by Daugherty and colleagues [58] was modified and adopted, as previously described [47]. Aneurysmal tissue was categorized independently by two blinded observers. There was no significant discordance in the designation by the two observers.

### 4.7. MMP-12 Inhibitor Administration

Alzet osmotic mini-pumps (model 1004; Charles River UK) were implanted subcutaneously in the interscapular region of relevant mice. The reservoir of each pump was preloaded with 100 µL of either sterile PBS (control) or a solution of the MMP-12 selective inhibitor RXP470.1 at a concentration of 4.6 mg/kg of body weight per day in sterile PBS (RXP470.1); this concentration and dosing regime was previously shown to be effective at suppressing atherosclerotic plaque progression in hypercholesterolemic Apoe^−/−^ mice [21]. Animals were treated simultaneously with both Ang II and RXP-470 for 3 or 4 weeks depending on the experimental design.

### 4.8. Termination

Animals were anaesthetized by intraperitoneal injection of sodium pentobarbitone (500 mg/kg), before exsanguination by perfusion via the heart with PBS at a constant pressure of 100 mmHg, with outflow through the incised jugular veins. This was followed by constant pressure perfusion with 10% formalin for at least 5 min.

### 4.9. Histological Analyses

Up to four vessel cross-sections were quantified per mouse, between 150 and 200 µm apart. Abdominal aortic sections were cut at 3 µm and subjected to the following histo- and immunohistochemical analyses. Quantitative analysis of elastin content and elastin breaks was performed using EVG-stained sections (four sections/vessel of interest/animal, 200 µm apart). The relative amount of elastin (which appears as black under light microscopy) was determined using a computerized image analysis program (Image Pro Plus, DataCell, Maidenhead, UK) and expressed as an average percentage of the aneurysmal artery, whilst the number of elastin breaks per section were counted and expressed as the average number of elastin breaks per section, averaged from four sections taken at 200 µm apart. Morphometric analysis was performed using a computerized image analysis program (Image Pro Plus, DataCell). The lengths of the internal and external elastic lamellae were recorded by image analysis. These were used to derive the total vessel area, lumen area, mean wall thickness, and maximum diameter by assuming them to be the circumferences of perfect circles. Quantitative analysis of fibrillar collagen content was performed using picrosirius red staining of sections viewed under polarized light. Qualitative analysis of the fiber thickness was assessed by delineating green and red fibers disposed under polarized light, as fiber color variation progresses from green to red proportionally to the increase of fiber thickness/ages, as described previously [47]. The relative amount of each fiber color was expressed as a percentage of the total amount of collagen within the area of interest.

### 4.10. Proteomics 

Abdominal aortae were harvested and snap-frozen for proteomics analysis. MirVana PARIS RNA and a Native Protein Purification Kit (ThermoFisher Scientific, Oxford, UK) were used to prepare protein lysates according to the manufacturer’s instructions. Briefly, abdominal aortae were placed into Precellys Ceramic kit 1.4 mm tubes (VWR, Leicester, UK) and 300 µL of cold disruption buffer was added to each tube. The samples were then subjected to 3 disruption cycles (10 s) using a Minilys Homogeniser (Bertin) at 4 °C. Protein samples (100 µg) were digested with trypsin (2.5 µg trypsin per 100 µg protein; 37 °C, overnight), labeled with Tandem Mass Tag (TMT) ten plex reagents according to the manufacturer’s protocol (ThermoFisher Scientific, Oxford, UK) and analyzed by LCMS using an Orbitrap Fusion Tribrid mass spectrometer running an SPS-MS3 acquisition, as described previously [59]. For proteomics data analysis (performed by the proteomics facility, BioMedical Sciences Building, University of Bristol), the raw data files were processed and quantified using Proteome Discoverer software v1.4 (ThermoFisher Scientific, Oxford, UK) and peptide sequences searched against the Uniprot Human database (134,169 sequences) using the SEQUEST algorithm. Peptide precursor mass tolerance was set at 10 ppm, and MS/MS tolerance was set at 0.6 Da. Searches were performed with full tryptic digestion and a maximum of 1 missed cleavage was allowed. The reverse database search option was enabled, and all peptide data was filtered to satisfy FDR of 5%. Only proteins that were detected in all samples were used in any analysis (accession numbers representing cDNA were excluded). Values for each of the proteins identified are presented as a ratio to the internal standard (a pool of all samples) and represent the median of the measured peptide(s) for each protein. High-fold differences (fold decrease/increase greater than 1.3) were determined and a log-2 transformation was applied. The significance between groups was determined by an unpaired *t*-test (treated compared with control), and −log 10 transformation was applied. Only significant fold changes (*p* < 0.05) were deemed of importance. The gene name, derived by mapping protein accession numbers within the Uniprot database from ‘UniprotKB AC/ID’ to ‘Gene name’, is frequently used synonymously with protein names throughout this paper. Protein (gene) names were inputted into Ingenuity Pathway Analysis (Qiagen) database alongside the STRING: functional protein association networks database (https://string-db.org accessed on 21 March 2024) to obtain regulatory and functional classification information to identify upstream regulators of the identified differentially expressed proteins, and determine enriched GO terms, versus a background list comprising all proteins detected during the proteomics analysis. A *p* value threshold of 0.05 was set.

### 4.11. Statistical Analyses 

Values are expressed as mean ± standard deviation (SD). Group values were compared using the computer program Prism (Version 10.2.0, GraphPad, La Jolla, CA, USA). For the comparison of group means, a check was first made for normal distribution. If this was passed, then an unpaired two sample two-tailed Student’s *t*-test was carried out. If the variances were significantly different, then a Welch’s correction test was used. Statistical differences between cells from the same preparation were analyzed by Student’s paired *t*-test. For the comparison of multiple groups, an analysis of variance (ANOVA) test was used, and a Student–Newman–Keuls multiple comparisons post hoc test was employed when statistical differences were detected. Contingency data (for example, aneurysm incidence) were analyzed by Fisher’s exact test. All in vivo and histological analyses were performed by two investigators in a blinded fashion.

## 5. Conclusions

Based upon our present results, we demonstrated that pharmacological inhibition of MMP-12 activity suppresses Ang II-induced AAA formation and progression in mice, through diminished macrophage accumulation and associated decreased elastin degradation (Figure 5). The findings presented here provide strong evidence demonstrating the deleterious role of aberrant MMP-12 activity in AAA development, alongside providing proof-of-concept evidence to motivate translational work upon MMP-12 inhibitor therapy in humans.

## Figures and Tables

**Figure 1 ijms-25-05809-f001:**
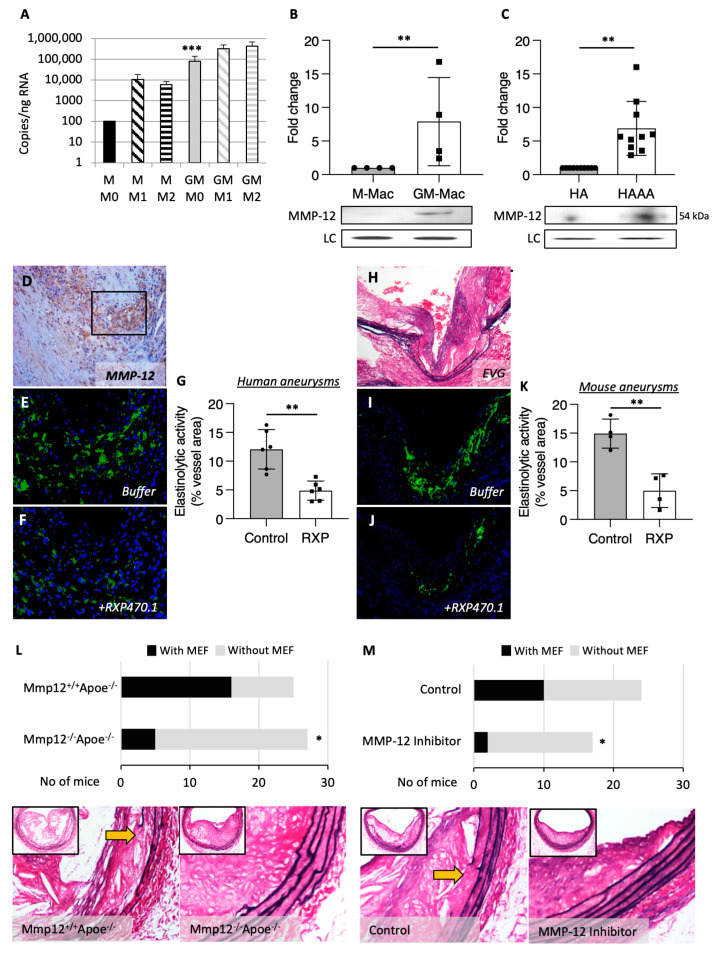
MMP-12 was elevated in pro-inflammatory macrophages and human AAA, and reduced MMP-12 expression/activity-retarded medial destruction. (**A**,**B**) MMP-12 mRNA and protein levels quantified by Q-PCR and Western blotting, respectively, in human monocyte-derived macrophages differentiated for 7 days with either M-CSF (M; 20 ng/mL) or GM-CSF (GM; 20 ng/mL) and subsequently activated with IFNγ and LPS (M1; 20 ng/mL and 100 ng/mL, respectively) or IL-4 (M2; 20 ng/mL), (*** *p* < 0.001 vs. M M0; ** *p* < 0.01 vs. GM-Mac; *n* = 6), LC denotes loading control. (**C**) MMP-12 protein expression in human healthy aorta (HA) and AAA (HAAA), (** *p* < 0.01; *n* = 10), LC denotes loading control. (**D**) Human AAAs containing abundant MMP-12-positive macrophages (brown color, scale bar: 100 µm), which are associated with elastinolytic activity as assessed by in situ zymography (green color; **E**), which can be retarded by a selective MMP-12 inhibitor (RXP470.1; (**F**)). Panels (**E**,**F**) are high power images from the area indicated by the black box in panel (**D**) (scale bar: 50 µm). (**G**) Quantification of elastinolytic activity in human AAA sections incubated with and without RXP470.1 (*n* = 6, two-tailed Student’s *t*-test, ** *p* < 0.01). (**H**) Mouse atherosclerotic AAA identified in elastin van Gieson (EVG)-stained sections where elastin fibers appear black, which is associated with elastinolytic activity as assessed by in situ zymography (green color; (**I**)) which can be retarded by a selective MMP-12 inhibitor (RXP470.1; (**J**)). Nuclei are counterstained with DAPI (blue), in panels (**E**,**F**,**I**,**J**). (**K**) Quantification of elastinolytic activity in AAAs from Ang II-infused Apoe-deficient mice incubated with and without RXP470.1 (*n* = 4, ** *p* < 0.01). (**L**) Representative images and analysis of atherosclerotic pseudo-aneurysms (PA) from Mmp12^+/+^Apoe^−/−^ and Mmp12^−/−^Apoe^−/−^ mice fed a high-fat diet for 8 weeks, within the brachiocephalic artery (scale bar: 50 µm). Elastin breaks (indicated by arrows) were identified in EVG-stained sections where elastin fibers appear black (*n* = 25–27, Fishers’ Exact test, * *p* < 0.05). (**M**) Representative images and analysis of atherosclerotic aneurysms for medial elastin fragmentation (MEF) from Apoe^−/−^ mice fed a high-fat diet for 12 weeks without or with MMP-12 inhibitor (RXP470.1) administration for the last 4 weeks, within the brachiocephalic artery (scale bar: 50 µm). Elastin breaks were identified in EVG-stained sections where elastin fibers appear black (*n* = 17–27, Fishers’ Exact test, * *p* < 0.05). All data are presented as mean ± SD.

**Figure 2 ijms-25-05809-f002:**
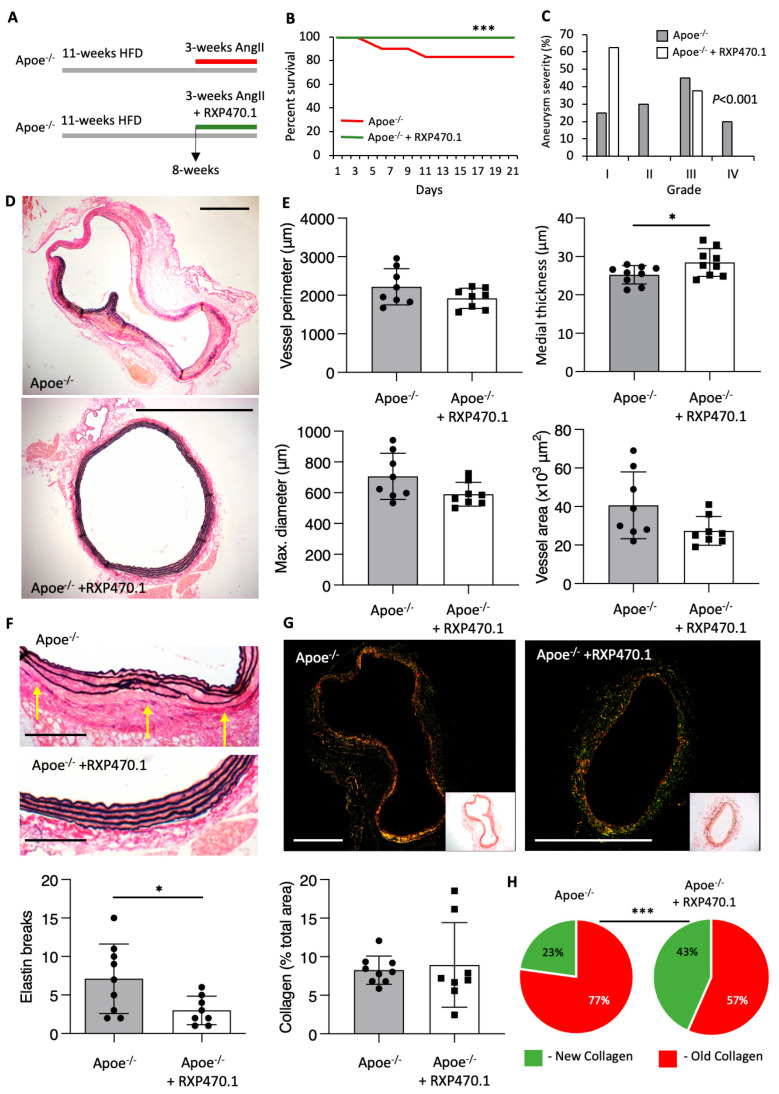
MMP-12 inhibition exerts protective effects on atherosclerotic AAA development. (**A**) Apoe^−/−^ mice were implanted with subcutaneous osmotic minipumps infusing Ang II (500 ng/kg/min) and were injected intraperitoneally with an MMP-12 inhibitor (RXP470.1) at 4.6 mg/kg or phosphate buffered saline (PBS) as a control, every 7 days for 3 weeks. (**B**) Kaplan–Meier plots of survival free from aneurysm rupture in 21-day Ang II-infused hypercholesterolemic Apoe^−/−^ mice, with and without RXP470.1 administration, *n* = 9/group, log-rank test, *** *p* < 0.001. (**C**) Quantification of aneurysm severity (increasing severity from stage I to IV) in control and RXP470.1-treated Apoe^−/−^ mice, *n* = 8–9/group, Fishers’ exact test, *** *p* < 0.001. (**D**,**E**) Representative images and morphological quantification of EVG-stained sections of abdominal aorta from control and RXP470.1-treated Apoe^−/−^ mice for vessel perimeter, maximal aortic diameter, medial thickness, and total vessel area (scale bar: 250 µm), *n* = 8–9/group, two-tailed Student’s *t*-test, * *p* < 0.05. (**F**) Representative images and quantification of elastin breaks (depicted by yellow arrows) in EVG-stained cross-sections of AAA from control and RXP470.1-treated Apoe^−/−^ mice (scale bar: 50 µm), *n* = 8–9/group, two-tailed Student’s *t*-test, * *p* < 0.05. (**G**,**H**) Representative images and total collagen content quantification of picrosirius red stained sections viewed under white light and linearly polarized light to show fibrillar collagen in AAA from control and RXP470.1-treated Apoe^−/−^ mice (scale bar: 250 µm), *n* = 8–9/group, two-tailed Student’s *t*-test, alongside associated qualitative analysis of new (green) and old (red) fibrillar collagen content, *n* = 8–9/group, Fishers’ exact test, *** *p* < 0.001. All data are presented as mean ± SD.

**Figure 3 ijms-25-05809-f003:**
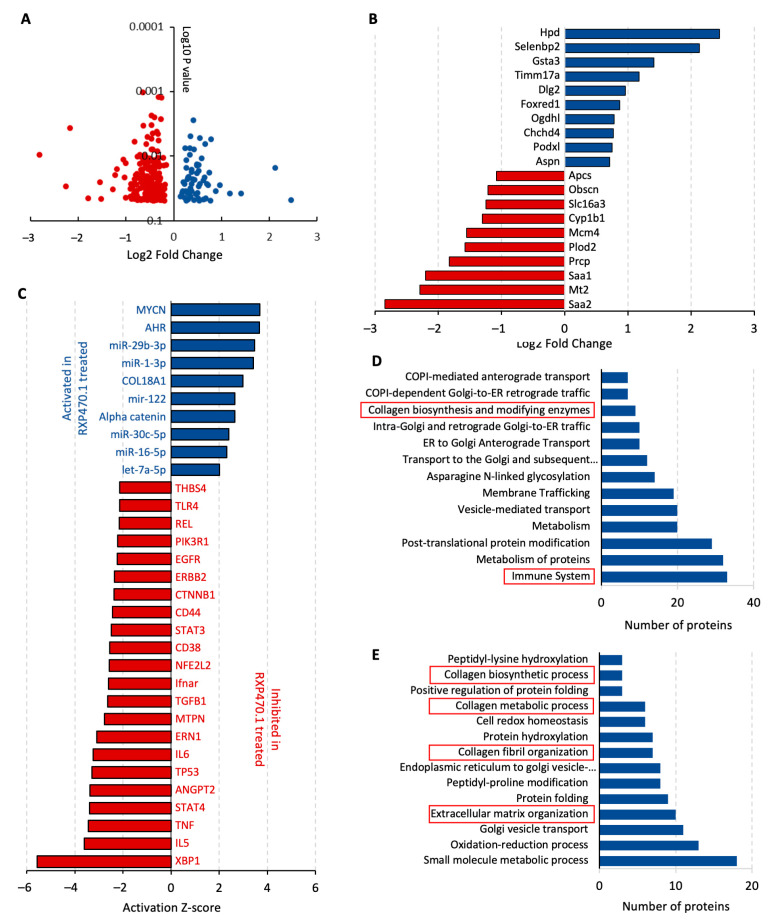
MMP-12 inhibition regulates ECM remodeling proteins and inflammatory pathways during early atherosclerotic AAA formation. Abdominal aortae were isolated from Apoe^−/−^ mice fed a high-fat diet for 11 weeks and infused with Ang II for the last 3 weeks of feeding, with or without RXP470.1 administration (*n* = 4/group) and subjected to proteomics analysis. (**A**) Volcano plot showing 266 proteins whose is expression is changed (* *p* < 0.05 and >1.5-fold change) between 21-day Ang II–infused hypercholesterolemic Apoe^−/−^ mice treated with and without RXP470.1, with red dots indicating proteins downregulated with RXP470.1 administration and blue dots indicating upregulated proteins. (**B**) Top ten upregulated (blue bars) and downregulated (red bars) proteins. (**C**) Deregulated upstream regulators whose targets were significantly overrepresented (* *p* < 0.05, Fisher’s exact test) among proteins exclusively affected by RXP470.1 administration. (**D**,**E**) Identification of numerous pathways deregulated (* *p* < 0.05, GSEA) by RXP470,1 administration, as determined by gene ontology (GO) and reactome enrichment analysis of differentially expressed proteins, respectively. Pertinent changed pathways are indicated by red boxes.

**Figure 4 ijms-25-05809-f004:**
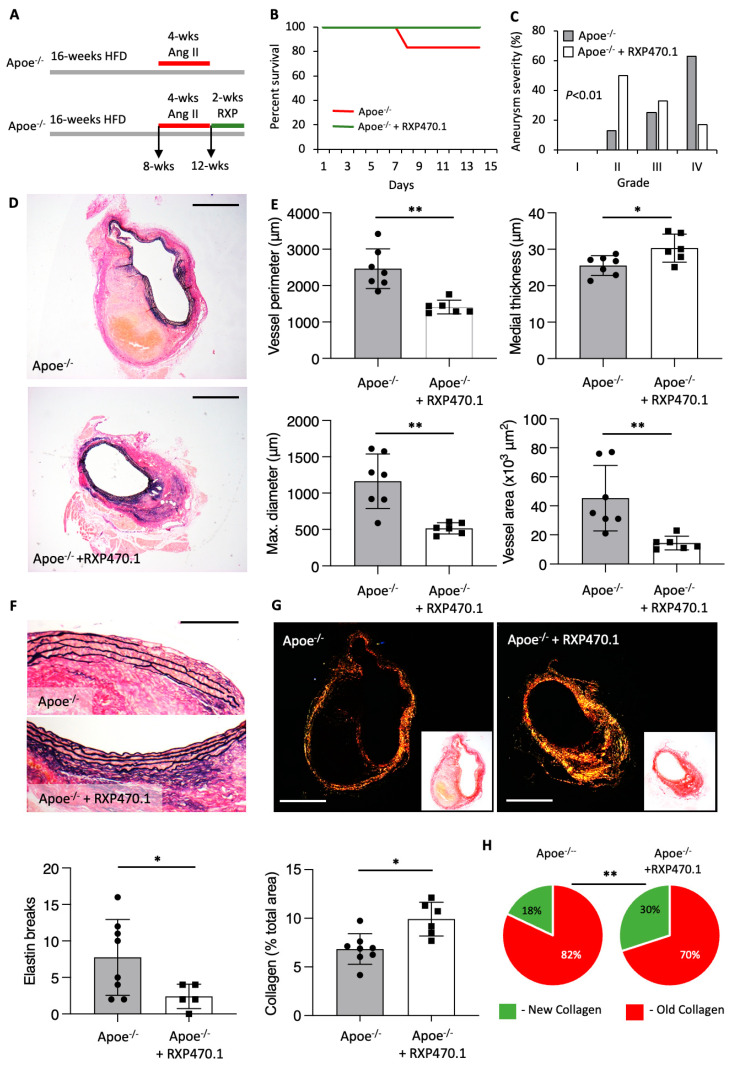
MMP-12 inhibition halts the progression of pre-existing atherosclerotic AAAs. (**A**) Apoe^−/−^ mice were implanted with subcutaneous osmotic minipumps infusing Ang II (500 ng/kg/min) and were injected intraperitoneally with an MMP-12 inhibitor (RXP470.1) at 4.6 mg/kg or PBS as a control, twice weekly for 2 weeks. (**B**) Kaplan–Meier plots of survival free from aneurysm rupture in 28 -day Ang II-infused hypercholesterolemic Apoe^−/−^ mice with and without RXP470.1 administration for two subsequent weeks, *n* = 7–8/group, log-rank test, * *p* < 0.05. (**C**) Quantification of aneurysm severity (increasing severity from stage I to IV) in control and RXP470.1-treated Apoe^−/−^ mice, *n* = 7–8/group, Fishers’ exact test, (**D**,**E**) Representative images and morphological quantification of EVG-stained sections of AA from control and RXP470.1-treated Apoe^−/−^ mice for vessel perimeter, maximal aortic diameter, medial thickness, and total vessel area (scale bar: 250 µm), *n* = 7–8/group, two-tailed Student’s *t*-test, ** *p* < 0.01, and * *p* < 0.05. (**F**) Representative images and quantification of elastin breaks in EVG-stained cross-sections of AAA from control and RXP470.1-treated Apoe^−/−^ mice (scale bar: 50 µm), *n* = 7–8/group, two-tailed Student’s *t*-test, * *p* < 0.05. (**G**,**H**) Representative images and total collagen content quantification of picrosirius red-stained sections viewed under white light and linearly polarized light to show fibrillar collagen in AAAs from control and RXP470.1-treated Apoe^−/−^ mice (scale bar: 250 µm), *n* = 7–8/group, two-tailed Student’s *t*-test, alongside associated qualitative analysis of new (green) and old (red) fibrillar collagen content, *n* = 7–8/group, Fishers’ Exact test, ** *p* < 0.01. All data are presented as mean ± SD.

**Figure 5 ijms-25-05809-f005:**
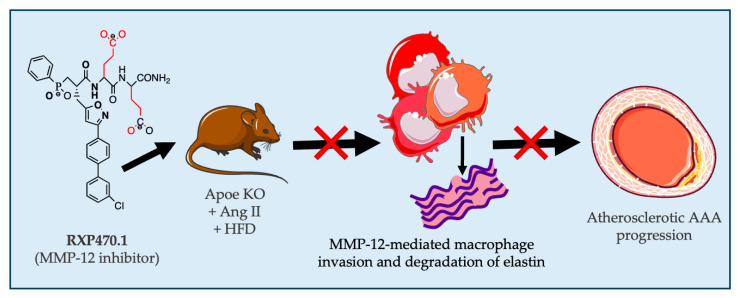
Graphical abstract demonstrating how pharmacological inhibition of macrophage-derived MMP-12 activity with RXP470.1 suppresses the progression of atherosclerosis AAA progression in Ang II-infused hypercholesterolmic mice, through diminished macrophage invasion and associated elastin degradation.

## Data Availability

The data that support the findings of this study are available from the corresponding author upon reasonable request. Some data may not be made available because of privacy or ethical restrictions.

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
