# Peer review of "Pharmacological Inhibition of MMP-12 Exerts Protective Effects on Angiotensin II-Induced Abdominal Aortic Aneurysms in Apolipoprotein E-Deficient Mice"

_ijms, 2024, doi:10.3390/ijms25115809_

Round 1

Reviewer 1 Report

Comments and Suggestions for Authors

Pharmacological Inhibition of MMP-12 Exerts Protective Effects on Angiotensin II-Induced Abdominal Aortic Aneurysms in Apolipoprotein E-Deficient Mice

Di Gregoli K et al

Human abdominal aortic aneurysms (AAAs) are characterized by increased activity of matrix metalloproteinases (MMP), including MMP-12, alongside macrophage accumulation and elastin degradation, in conjunction with superimposed atherosclerosis. Previous genetic ablation studies have proposed contradictory roles for MMP-12 in AAA development. In this study, Di Gregoli et al aimed to elucidate if pharmacological inhibition of MMP-12 activity with a phosphinic peptide inhibitor, protects from AAA formation and progression in Ang II-infused Apoe-/- mice. Complimentary studies were conducted in a human ex vivo model of early aneurysm development. Administration of an MMP-12 inhibitor (RXP470.1) protected hypercholesteroleamic Apoe-/- mice from angiotensin (Ang) II-induced AAA formation and rupture-related deaths, associated with diminished medial thinning and elastin fragmentation alongside increased collagen deposition. Proteomic analysis confirmed a beneficial effect of MMP-12 inhibition on extracellular matrix remodelling proteins combined with inflammatory pathways. Furthermore, RXP470.1 treatment of mice with pre-existing AAAs exerted beneficial effects as observed through suppressed aortic dilation and rupture, medial thinning, and elastin destruction. Their findings indicate that pharmacological inhibition of MMP-12 activity retards AAA progression and improves survival in mice providing proof-of-concept evidence to motivate translational work for MMP-12 inhibitor therapy in humans.

Major Comments:

The authors have performed an incredible amount of work in this study. The only issues I have with this manuscript are that the authors need to: (1) mention the role of heme oxygenase-1 in AAA development; and (2) express their data as mean±SD and not SEM, errors bars would then be more reflective of the true variance of the results. Figures need to be revised as a result. Also, would like to see an explanation as to why no Stage II was observed but Stage III after treatment with RXP470.1. Also, there are a few errors that need to be addressed (see detailed comments below). But all in all, great work here.

Minor Comments:

Abstract:

Pg. 1,   line 9: change “AAA” to “AAAs”; “characterised” to “characterized”.

            line 16: insert “angiotensin” after “from”; change “Ang” to “(Ang)”.

            line 18: change “analysis” to “analyses”.

Introduction:

Pg. 1,   line 29: change “aetiology” to “etiology”.

            line 30: insert “can” after “aneurysms”.

            line 31: change “AAA” to “AAAs”; what is meant by “demonstrate prevalence”.

            line 32: change “harbour” to “harbor”.

            line 35: change “stabilising” to “stabilizing”.

            line 37: change “standardised” to “standardized”.

            line 40: change “characterised” to “characterized”.

            line 42: define “VSMC”; change “existence” to “presence”.

Pg. 2,   line 45: define “ECM”; what is meant by “distinguish”?” from what?

line 46: change “characterise” to “characterize”.

            lines 47 to 48: delete “abdominal…aneurysms”.

line 48: change “(AAA)” to “AAAs”; “AAA” to “AAAs”.

line 51: change “AAA” to “AAAs”.

line 54: change “to” to “with”.

line 56: change “to” to “with”; “AAA” to “AAAs”.

line 58: change “aneurysm” to “aneurysms”.

lines 60 and 61: change “AAA” to “AAAs”.

line 62: define “MMPs”.

line 63: change “AAA” to “AAAs”.

line 64: change “Matrix…)” to “MMPs”.

line 67: change “avenue” to “strategy”.

line 70: change “AAA” to “AAAs”.

line 74: change “aorta” to “aortae”.

line 78: change “observations” to “findings”.

line 81: define “Ang”; why is “Apoe-/-” italicized?

line 83: change “AAA” to “AAAs”.

Results

Pg 2     line 87: change “AAA” to “AAAs”.

            line 92: change “level” to “levels” and “polarised” to “polarized”.

lines 93 and 94: please define “GM-CSF”; “M-CSF”; IFN”; “LPS”; “IL-4”.

Pg 3     line 96: change “abdominal…aneurysms” to “AAAs”; “to” to “with”.

            line 97: insert “-” after “12”; change “AAA” to “AAAs”.

            line 100: change “AAA” to “AAAs”; “Figure” to “Figures”.

            line 102: change “localised” to “localized”.

            line 104: change “Figure” to “Figures”.

            line 106: change “to” to “with”.

            line 107: insert “(MEF)” before “was”.

line 109: change “implied” to “suggested”; “maybe” to “may be”.

Pg 4     line 112: change “is” to “was”.

            line 113: change “retards” to “retarded”.

            line 115: change “ml” to “mL” X 2.

            line 118: change “AAA” to “AAAs”.

line 119: insert “-” after “12”; change “colour” to “color”.

line 120: change “colour” to “color”.

Pg 5     line 124: change “AAA” to “AAAs”; define “EVG”; insert “-“ after “EVG”.

line 125: change “colour” to “color”.

            line 127: insert “,” after “I”.

            line 131: insert “-“ after “EVG”.

            line 133: delete “pseudo…(“ and “)”.

line 135: insert “-“ after “EVG”.

line 136: please express data as “mean±SD“ and not “mean±SEM”.

            line 138: change “abdominal…aneurysm” to “AAA”.

            line 139: change “hypercholesterolaemic” to “hypercholesterolemic”.

            line 140: change “within” to “with”.

            line 144: change “to” to “with”.

            line 145: interesting that no Stage II but Stage III after treatment… why so?

line 146: insert “-“ after “diameter”.

line 148: change “Figure” to “Figures”.

            line 152: change “fibres” to “fibers”.

            line 153: change “signifying” to “demonstrating”.

            line 154: delete “imparted”.

Pg 6     line 161: define “PBS”; change “curves” to “plots”.

line 164: delete second “stage”.

            line 166: delete “elastin…(“ and “)”.

Pg 7     line 176: please express data as ”mean±SD”.

            line 177: change “extracellular matrix” to “ECM”.

            lines 180 to 181: change “obtain…into” to “elucidate”.

            line 182: insert “-” after “differentially”.

            line 183: change “p” to “P”.

            line 184: delete “-” X 2.

line 185: change “AAA” to “AAAs”.

            lines 186 and 187: delete “-”.

            line 189: insert “,” after “responses”; insert “-” after “Differentially”.

            lines 191, 201, 203: change “p” to “P”.

line 206: define “FDR”.

Pg. 8    line 217: change “extracellular matrix” to “ECM”.

line 224: delete “-” X 3.

            line 234: delete “-”; insert “of” before “Ang”.

            line 235: change “harbouring” to “harboring”; delete “-”.

Pg. 9    line 239: change “to” to “with”.

line 241 and 244: change “Figure” to “Figures”.

            line 252: delete first “-”; change “curves” to “plots”.

Pg. 10  line 255: delete second “stage” to “with”.

line 257: delete “elastin…(” and “)”;  change “abdominal aorta” to “AA”.

line 263: insert “-” after “red”.

line 264: change “AAA” to “AAAs”.

line 267: please express data as ”mean±SD”.

Discussion:

Pg 10   line 271: change “utilising” to “utilizing”.

            line 289: change “to” to “with”; “aorta” to “aortae”.

line 291: change “aortic aneurysms” to “AAs”.

            line 296: delete “-”. 

            line 302: change “normocholesterolaemic” to “normocholesterolemic”.

line 306: change “characterises” to “characterizes”; aortic aneurysms” to “AAs”; “utilises” to “utilizes”.

line 308: change “hypercholesterolaemic” to “hypercholesterolemic”.

Pg 11   line 318 and 324: delete “-”.

line 325: change “hypercholesterolaemic” to “hypercholesterolemic”.

            line 334: chsange “fibres” to “fibers”.

            line 336: change “Supportingly” to “In support”.

            line 348: change “hypercholesterolaemic” to “hypercholesterolemic”.

line 357: change “harbouring” to “harboring”.

Pg 12   line 364: change “hypercholesterolaemic” to “hypercholesterolemic”.

line 378: change “utilising” to “utilizing”.

line 379: change “for” to “of”.

line 380: change “favourable” to “favorable”.

line 382: change “model” to “models”.

line 384: change “content” to “contents”.

line 390: change “the aorta” to “aortae”.

line 391: change “provide” to “provided”.

line 392: change “deployed” to “taken”.

line 393: change “harbouring” to “harboring”.

Materials and Methods:

Pg. 12  line 403: define “RPMI”.

lines 404 to 405: delete “macrophage…(” and “)”.

lines 405 to 406: delete “granulocyte…(” and “)”.

            line 409: delete “interferon (”; “lipopolysaccharide (”.

            line 410: delete “interleukin 4 (”.

Pg 13   line 435: define “HRP”.

            line 442: change “aortic aneurysms” to “AAs”.

            line 446: change “Cl2” to “Cl2”.

line 455: delete “the”.

Pg 14   line 463: delete “Abdominal…Aneurysm (“ and “)”.

line 484: change “µl” to “µL”; “phosphate…saline” to “PBS”.

line 486: change “phosphate…saline” to “PBS”.

            line 487: delete “-”.

line 490: insert “space” after “500”.

line 493: change “Analysis” to “Analyses”.

line 494: insert “space” after “200”.

line 495: insert “space” after “3”.

            line 496: delete “-”.

line 497: change “elastin…Gieson” to “EVG-”.

lines 498 and 503: insert “space” after “200”.

line 508: change “polarised” to “polarized”.

line 509: change “fibre” to “fiber”.

line 510: change “polarised” to “polarized”; “fibre colour” to “fiber color”.

Pg 15   line 512: change “fibre colour” to “fiber color”.

line 515: change “aortas” to “aortae”; “proteomic” to “proteomics”.

line 517: change “aortas” to “aortae”.

line 518: change “µl” to “µL”.

lines 520 and 521: insert “space” after “100”.

line 521: insert “space” after “2.5”.

line 515: change “proteomic” to “proteomics”.

line 529: insert “space” after “10” and “0.6”.

line 532: delete “false…(” and “)”.

line 538: define “VSD” and “TOF”.

line 539: change “p” to “P”.

lines 545 to 546: delete “gene…(” and “)”.

line 548: change “Analysis” to “Analyses”.

line 549: please show data as “mean±SD”.

line 560: there are no Bland-Altman plots shown, should be Kaplan-Meier plots right?

Figure 5 Legend: 

Pg 16   line 1: delete “-” between “macro” and “phage”.

line 2: change “abdominal…aneurysm” to “AAA”.

line 3: change “hypercholesterolaemic” to “hypercholesterolemic”.

References:

Fix these references: formatting of titles is sentence case #1, 7, 10, 12, 15, 19, 20, 21, 25, 27, 28, 29, 30, 32, 33, 34, 37, 43, and 44.

Comments on the Quality of English Language

needs moderate improvement, see detailed comments above.

Reviewer 2 Report

Comments and Suggestions for Authors

Abstract and introduction: the authors gave a good overview about the role of MMP-12 and macrophages in the development of abdominal aortic aneurism (AAA) together with the genetic predisposition to the disease.

Materials and methods: well written, easy to follow. The authors described all methods and materials used in the study. Although IFN-γ was used to differentiate macrophages, the “gamma” was only used in one sentence, otherwise it was named as IFN. It is suggested to name it as INF-γ.

English language: appropriate wording was used, no big issues were detected. Some typos are existing, please check the manuscript.

Figures:  the manuscript contains 5 figures. In Materials and Methods it is written that 20 ng/mL M-CSF or GM-CSF was used for differentiation, however, in the figure legend (Figure 1) 25 ng/mL was written. Please give the accurate amount of the factors used. The amount of other factors (i.e. LPS) are only given at Materials and Methods. Please add to figure legend. Figure legend at Figure 5 contains a typo. Please correct it.

Results: Wide variety of experiments were performed in order to validate that the inhibition of MMP-12 blocks the progression of AAA. Experiments and results are easy to follow, figures support the understanding of the results. To support translational research, experiments were performed on mice and human aortic tissue.

Discussion and conclusion: The authors took results in a wider literature-based context. They also highlighted the potential limitation of using MMP-12 inhibitors as a therapeutic option.

Taking into account that the manuscript is well written, it is easy to follow and understand, only minor typos are present, the manuscript is suggested to be accepted in the current form.

Comments on the Quality of English Language

No  major issues were detected. Please check typos, i.e.: figure legend at Figure 5.

Reviewer 3 Report

Comments and Suggestions for Authors

The study seemed to be well done and the findings were comparatively firm.

1.      The progression of AAA includes acute and/or chronic phase. Is MMP-12 involved in acute dilation, rupture, and dissection? With caution of the phase, the conclusion should be described.

2.      Introduction (row 51); in prior studies, do some genetic components associated with AAA include MMP-12-related genes? It could be mentioned here.

3.      Introduction (row 51); in prior studies, lipoprotein (i.e., Lp(a)) and inflammatory markers are reported as several risk factors of AAA. These could be described here.

4.      Row 481; the inhibitor dose and dosage; how was that determined (for instance, prior literature?).

5.      The expression, AAA and AAAs, were mixed in the text. It is confusing.

6.      The expression, Apoe-/- and apoe-/-(superscript), were mixed in the text.

7.      Row 48; AAA was again abbreviated (it was already abbreviated).

8.      Row 94; after INF, a large space happened.

9.      Row 389, before conversely, a large space happened.

10.  Row 453, before Mice, ‘. (period)’ was seen.

11.  Row 463, AAA was again abbreviated (it was already abbreviated).

12.  Row 488, before Termination, ‘. (period)’ was seen.

13.  Row 554; Students paired t-test; student’s or student was right.

14.  Row 558; in vivo can be written in Italic.

15.  This is a simple question; are there any treatments of MMP-12 by lifestyles or generic popular drugs? If any, that might be more added to the discussion.

Comments on the Quality of English Language

Please check it again by native speakers.

Author Response

Rebuttal

Reviewer #3

We are grateful to the reviewer for their fair appraisal of the manuscript, and the constructive comments. Please find a point-by-point clarification of the associated revisions.

  1. The progression of AAA includes acute and/or chronic phase. Is MMP-12 involved in acute dilation, rupture, and dissection? With caution of the phase, the conclusion should be described.

Rather than the phase, we believe our findings highlight the discrepancy between atherosclerotic AAAs (as exist in humans) and non-atherosclerotic AAAs (commonly observed in mouse models that do not incorporate hypercholesterolemia). Accordingly, we have revised the conclusion to reinforce this tenet while also integrating some of the points you make in your comment. The revised sentences read as such:

“In conclusion, our current findings confirm a deleterious role for aberrant MMP-12 activity in atherosclerotic aneurysm formation and rupture as demonstrated by the protective effect afforded on the formation and progression of established AAAs through MMP-12 inhibition with RXP470.1. Conversely, previous evidence suggests in the absence of atherosclerosis and associated inflammation, MMP-12 protects the integrity of the aortae and prevents acute dilation and subsequent aortic dissection.”

  1. Introduction (row 51); in prior studies, do some genetic components associated with AAA include MMP-12-related genes? It could be mentioned here.

We have added the following sentence:

Although earlier genetic studies failed to identify an association for MMP-12 single nucleotide polymorphisms with AAAs [22], more recent genome-wide association studies have supported a genetic link between MMP-12 and AAA risk [23].”

  1. Introduction (row 51); in prior studies, lipoprotein (i.e., Lp(a)) and inflammatory markers are reported as several risk factors of AAA. These could be described here.

We have added the following sentence:

Relatedly, indicators of dysfunctional lipid metabolism (including raised circulating low density lipoprotein-cholesterol (LDL-C) or lipoprotein(a) levels) and systemic inflammation (such as interleukin (IL)-6 and interferon-g (IFNg) concentrations) are associated with the presentation of AAAs [7, 8].”

  1. Row 554; Students paired t-test; student’s or student was right.

Checked and all changed to ‘student’s’.

  1. This is a simple question; are there any treatments of MMP-12 by lifestyles or generic popular drugs? If any, that might be more added to the discussion.

We have added the following sentences to the discussion to address this comment:

“It is plausible that existing cardiovascular therapeutics utilized to treat associated risk factors or recommended lifestyle changes may directly alter MMP-12 levels and/or activity, and therefore be effective for the treatment of AAAs. While plasma levels of MMP-12 have been shown to be increased in individuals with type II diabetes mellitus, with elevated systolic blood pressure, or with increasing age [49], decreased circulating levels are observed in cardiovascular patients receiving anti-hypertensive drugs [50] or statin therapy [51]. Experimentally, the LDL-C lowering drug atorvastatin was demonstrated to inhibit the development of AAAs in a rat elastase-induced model, an effect attributed to its anti-inflammatory actions including suppression of macrophage accumulation and associated reduced MMP-12 expression [52].”

Round 2

Reviewer 3 Report

Comments and Suggestions for Authors

The paper was much improved.